# SELF-LEARNING COMPOSITIONAL REPRESENTATIONS FOR ZERO-SHOT CHINESE CHARACTER RECOGNITION

## ABSTRACT

Chinese character recognition has been a longstanding research topic and remains essential in visual tasks like ancient manuscript recognition. Chinese character recognition faces numerous challenges, particularly the issue of zero-shot characters. Existing Chinese zero-shot character recognition methods primarily focus on the radical or stroke decomposition. However, radical-based methods still struggle to solve zero-shot radicals, while stroke-based ones are hard to perceive fine-grained information. Besides, previous methods can hardly generalize to characters of other languages. In this paper, we propose a novel **S**elf-learning **C**ompositional **R**epresentation method for zero-shot **C**hinese **C**haracter **R**ecognition (**SCR-CCR**). SCR-CCR learns compositional components automatically from the data, which are not aligned with human-defined radical or stroke decomposition methods. SCR-CCR follows the pretraining-inference paradigm. First, we train a Character Slot Attention (ChSA) via pure feature reconstruction loss to parse appropriate components from character images. Then we recognize zero-shot characters without finetuning or retraining in the inference stage by comparing components between input and example images. To evaluate the proposed method, we conduct experiments of zero-shot character recognition. The experiments illustrate that SCR-CCR outperforms previous methods in most cases of character and radical zero-shot settings. In particular, visualization experiments indicate that the components learned by SCR-CCR reflect the structure of characters in an interpretable way, and can be used to recognize Japanese and Korean characters.

## 1 INTRODUCTION

Optical Character Recognition (OCR) plays a crucial role in various downstream tasks, such as document understanding (Francois et al., 2022; Singh & Sachan, 2018) and traffic sign recognition (Jain & Gianchandani, 2018; Greenhalgh & Mirmehdi, 2014). Thus, this field continues to attract the attention of researchers. Unlike Latin characters, Chinese characters possess complex internal structures. Consequently, the multitude of Chinese character categories often leads to the prevalence of zero-shot learning problems in practical applications (Yu et al., 2023).

As Figure 1 illustrates, previous zero-shot Chinese Character Recognition (CCR) methods can be broadly categorized into character-based, radical-based and stroke-based approaches. To solve the zero-shot problem, the character-based approach (Li et al., 2020; Xiao et al., 2019) extracts monolithic representation for images and typically requires additional printed character images during training. Differently, the radical- or stroke-based approaches (Wang et al., 2019; 2018; Chen et al., 2021) recognize Chinese characters through radical and stroke decomposition, which may cost considerable inference time due to the existence of auto-regressive decoders. Recently, based on CLIP (Radford et al., 2021), Yu et al. (2023) proposed an efficient image-IDS matching method for zero-shot CCR. Although existing methods have achieved certain performance improvements in zero-shot CCR, the human-defined representations used in these methods may lack the flexibility to adapt to different scenarios and have poor generalization in practical applications.

Compositionality is a fundamental way in which humans understand and interpret the world (Lake et al., 2017). In contrast to monolithic representations of entire scenes, compositional representations describe the visual world by discovering objects in scenes, capturing attributes of objects, and

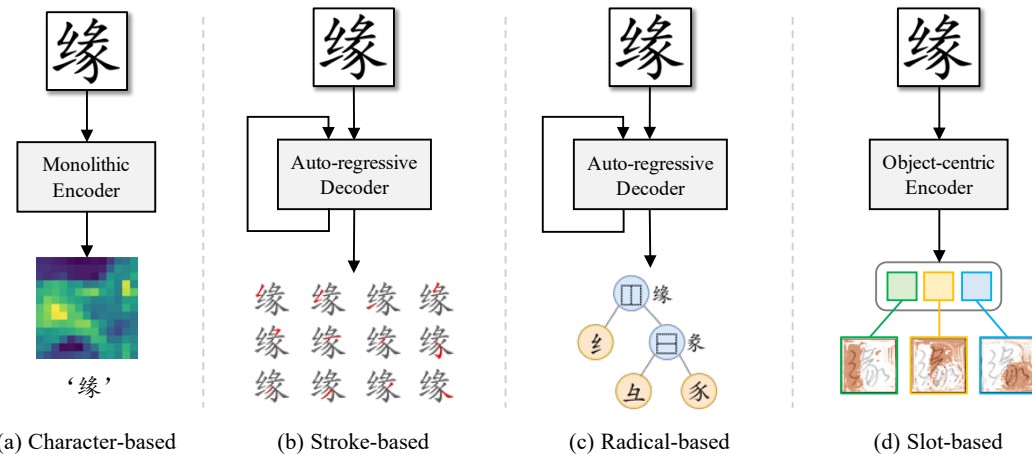

(a) Character-based  (b) Stroke-based  (c) Radical-based  (d) Slot-based

Figure 1: **Different representation methods of Chinese character recognition.** (a) displays character-based methods using monolithic representations to predict character labels; (b) and (c) are stroke-based and radical-based methods using auto-regressive decoders to predict human-defined strokes and radicals; (d) indicates the proposed SCR-CCR that can automatically decompose characters into object slots.

abstracting relationships between objects (Singh et al., 2022; Seitzer et al., 2022). As a typical compositional representation, object-centric representation is crucial for understanding visual scenes and enhancing generalization capabilities in scenes with novel objects or combinations of objects (Locatello et al., 2020; Dedhia et al., 2023). For example, if one can recognize a *car* and a *tree* as two independent objects, it can understand a new scene where a *car* parks next to a *tree*, even though it has never seen such combination. In zero-shot CCR, we can decompose unseen characters into acquired objects, and transform the task of character recognition into comparing object sequences (Lake et al., 2011; 2015). Although radical-based or stroke-based approaches are similar in the motivation of character decomposition, the object-centric representations are automatically learned from data and can handle different types of data without annotations of human-defined radical or stroke categories.

Inspired by the compositionality of visual scenes, we propose a novel **S**elf-learning **C**ompositional **R**epresentation method for zero-shot **C**hinese **C**haracter **R**ecognition (**SCR-CCR**). As Figure 1d shows, SCR-CCR parses slots (*i.e.*, compositional objects) from Chinese characters automatically, which are not aligned with human-defined structures such as radicals and strokes, allowing it to generalize effectively to unseen characters in a zero-shot setting. SCR-CCR realizes zero-shot CCR via a pretraining-inference paradigm. In the first pretraining stage, we train an encoder, decoder, and Character Slot Attention (ChSA) to parse appropriate slots from input character images by reconstructing features of a frozen pre-trained encoder (Locatello et al., 2020). In the second inference stage, SCR-CCR recognizes zero-shot characters without finetuning by matching slots of input and example images. We conduct experiments of character and radical zero-shot CCR. SCR-CCR outperforms previous methods on all datasets in both zero-shot settings and surpasses previous methods by about 50% in the radical zero-shot setting. Visualization experiments indicate that the slots learned by SCR-CCR can reflect the structure of Chinese characters in an interpretable way. In particular, trained with only Chinese characters, SCR-CCR can recognize Japanese and Korean characters, achieving an accuracy of 89% and 62%.

## 2 RELATED WORKS

### 2.1 ZERO-SHOT CHINESE CHARACTER RECOGNITION

Due to the significantly larger number of Chinese characters compared to Latin characters, character recognition in Chinese inevitably encounters zero-shot problems, *i.e.*, the characters in the test set are

excluded in the training set. Early works in Chinese character recognition can be broadly categorized into three types: character-based, radical-based, and stroke-based approaches.

**Character-based.** Before the era of deep learning, the character-based methods usually utilize the hand-crafted features to represent Chinese characters (Jin et al., 2001; Su & Wang, 2003; Chang, 2006). With deep learning achieving a great success, MCDNN (Cireşan & Meier, 2015) takes the first attempt to use CNN for extracting robust features of Chinese characters while approaching the human performance on handwritten CCR in the ICDAR 2013 competition (Yin et al., 2013). Although the character-based methods, treating each character as one class, have a higher time efficiency, they are prone to suffer from the character zero-shot problem in practice.

**Radical-based.** To solve the character zero-shot problem, some methods propose to predict the radical sequence of the input character image. In Wang et al. (2018), character images are first fed into a DenseNet-based encoder (Huang et al., 2017) to extract the character features, which are subsequently decoded into the corresponding radical sequences through an attention-based decoder. FewShotRAN (Wang et al., 2019) proposes a radical aggregation module to introduce the deep prototype learning for robust radical feature representation. These radical-based methods can indeed alleviate the character zero-shot problem to a certain extent, but the prediction of radical sequences takes longer time than the character-based methods. Although HDE (Cao et al., 2020) adopts a matching-based method to avoid the time-consuming radical sequence prediction, this method needs to manually design a unique vector for each Chinese character. Meanwhile, it does not achieve ideal performance in the zero-shot settings.

**Stroke-based.** To fundamentally solve the zero-shot problem, some methods decompose Chinese characters into stroke sequences. The early stroke-based methods usually extract strokes by traditional strategies. For example, in Kim et al. (1999), the authors employed mathematical morphology to extract each stroke in characters. The proposed method in (Liu et al., 2001) describes each Chinese character as an attributed relational graph. Recently, a deep-learning-based method (Chen et al., 2021) is proposed to decompose each Chinese character into a sequence of strokes and employs a feature-matching strategy to solve the one-to-many problem (*i.e.*, there is a one-to-many relationship between stroke sequences and Chinese characters). This stroke-based method can indeed alleviate the zero-shot problem and achieve higher performance than radical-based methods. However, it costs more time in inference, resulting from that the predicted stroke sequences of Chinese characters are longer than the corresponding radical sequences.

Recently, Yu et al. (2023) introduced CCR-CLIP, which aligns character images with their radical sequences to recognize zero-shot characters, achieving comparable inference efficiency with the character-based approach. All previous methods focus on learning Chinese character features through human-defined representations but struggle to achieve high generalization capabilities.

## 2.2 OBJECT-CENTRIC REPRESENTATION LEARNING

Object-centric representation methods interpret the world in terms of objects and their relationships. They capture structured representations that are more interpretable, compositional, and generalizable, which has become increasingly popular in computer vision, as it aligns with how humans perceive and interact with the world. One class of models extracts object-centric representations with feedforward processes. For example, SPACE and GNM (Lin et al., 2020; Jiang & Ahn, 2020) attempt to divide images into small patches for parallel computation while modeling layouts of scenes. Another class of models initializes and updates object-centric representations by iterative processes (Greff et al., 2017; 2019; Emami et al., 2021). A representative method is Slot Attention, which assigns visual features to initialized slots via iterative cross-attention mechanism (Locatello et al., 2020). Based on Slot Attention, many methods have been proposed to improve the quality of object-centric representations in different scenarios (Seitzer et al., 2023). Recently, some models have aimed at parsing object-centric scene representations in videos. SAVi++ (Elsayed et al., 2022) uses Slot Attention to extract a set of temporally consistent latent variables while discovering and segmenting objects with additional visual cues of the first video frame. STEVE (Singh et al., 2022) combines the transformer-based decoder of SLATE (Singh et al., 2021) with a standard slot-level recurrence module to extract object-centric representations.

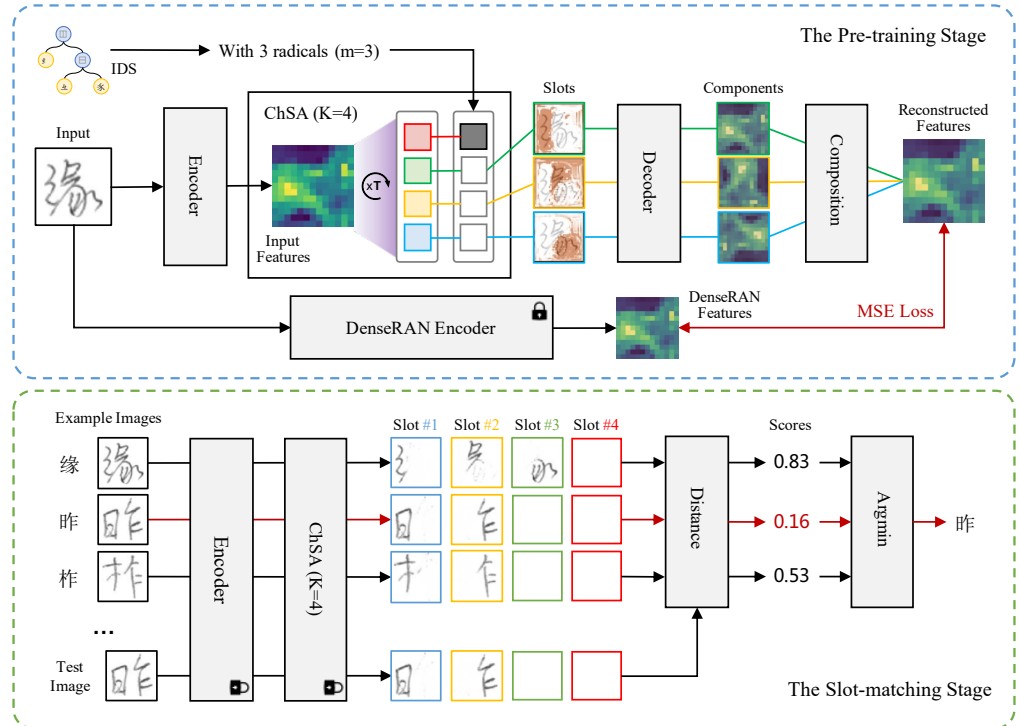

Figure 2: **An overview of the two stages in SCR-CCR.** SCR-CCR consists of a trainable encoder, ChSA and decoder. In the pre-training stage, SCR-CCR encodes image features, extracts slots, and decodes slots back to features. The training objective is to reduce the difference between the features reconstructed by SCR-CCR and the features of a frozen teacher encoder. In the slot-matching stage, SCR-CCR uses the pre-trained encoder and ChSA to extract slots from the test image and example images, assigning a category to the test image by comparing their slots.

## 3 METHODOLOGY

In this paper, we propose SCR-CCR, an object-centric representation method for CCR. As shown in Figure 2, SCR-CCR recognizes characters through two separate stages: the pre-training stage and the slot-matching stage. In the pre-training stage, we train an encoder, ChSA, and decoder that can extract slots (*i.e.*, object-centric representations) from the input character images. And in the slot-matching stage, we use the pre-trained ChSA to extract slots from the input and example images, and assign a category to the input by comparing their component slots.

### 3.1 PROBLEM SETTING

Given an input character image, a CCR model is required to provide the class of the input. In most cases, the training and testing splits of datasets have similar distributions of characters. However, all testing samples will not appear in the training stage in a more challenging zero-shot setting. The key to zero-shot character recognition is transferring the ability of character recognition to novel cases, which are not available in the model training. In the paper, a sample is a tuple $(\boldsymbol{X}, y)$ where $\boldsymbol{X}$ is the input character image and $y$ is the corresponding input type. We keep the training charset and the testing charset disjoint for the zero-shot setting.

### 3.2 SLOT ATTENTION

Slot Attention (Locatello et al., 2020) maps a scene to a group of slots to capture the representations of objects independently. This mechanism effectively extracts object-centric representations and can even automatically discover individual objects in a scene in an unsupervised setting. The core idea

is to iteratively assign the features of the input image with each slot through a specific attention mechanism.

**Input Feature Encoding**. First, the input image $\boldsymbol{X}$ is encoded into a 2D feature matrix $\boldsymbol{F} \in \mathbb{R}^{N \times D}$ through a feature extractor, where $N$ is the number of input features (*e.g.*, spatial locations or pixels), and $D$ is the dimensionality of each feature.

**Slot Initialization**. A set of slots $\boldsymbol{S} \in \mathbb{R}^{K \times D}$ is initialized randomly or with learnable parameters, where $\boldsymbol{S} = \{\boldsymbol{s}_1, ..., \boldsymbol{s}_K\}$, $K$ is the number of slots (*i.e.*, the maximum number of objects to extract), and each slot is a $D$-dimensional vector.

**Attention Computation and Slot Update**. In each iteration, the slots act as queries, while the input features act as keys and values. The matching is performed through an attention mechanism. First, compute the queries, keys, and values:

$$\boldsymbol{Q} = \boldsymbol{W}_q \boldsymbol{S} \in \mathbb{R}^{K \times D_q}, \quad \boldsymbol{K} = \boldsymbol{W}_k \boldsymbol{F} \in \mathbb{R}^{N \times D_k}, \quad \boldsymbol{V} = \boldsymbol{W}_v \boldsymbol{F} \in \mathbb{R}^{N \times D_v}, \quad (1)$$

where $\boldsymbol{W}_q$, $\boldsymbol{W}_k$, and $\boldsymbol{W}_v$ are linear transformation weight matrices that project the slots and input features into different dimensional spaces of $D_q$, $D_k$, and $D_v$, respectively. Slot Attention computes the attention logits by measuring the similarity between the slot queries and the feature keys, and normalizes the logits to h prevent ignoring parts of the input features:

$$\boldsymbol{A}_{ij} = \frac{e^{\boldsymbol{\Phi}_{ij}}}{\sum_l e^{\boldsymbol{\Phi}_{lj}}}, \quad \text{where } \boldsymbol{\Phi} = \frac{\boldsymbol{Q}\boldsymbol{K}^\top}{\sqrt{D_k}} \in \mathbb{R}^{K \times N}. \quad (2)$$

Slot Attention aggregates the input values to their assigned slots by a weighted mean operation:

$$\boldsymbol{U} = \boldsymbol{W}\boldsymbol{V} \in \mathbb{R}^{K \times D_v}, \quad \text{where } \boldsymbol{W}_{ij} = \frac{\boldsymbol{A}_{ij}}{\sum_l \boldsymbol{A}_{il}}. \quad (3)$$

Slot Attention use the aggregated values $\boldsymbol{U}$ to updates the slots:

$$\boldsymbol{S}_{\text{new}} = \text{GRU}(\boldsymbol{S}, \boldsymbol{U}). \quad (4)$$

The Gated Recurrent Unit fuses the newly extracted information with the previous slot states. The above process is repeated over multiple iterations to update the slots through the attention mechanism, allowing them to gradually focus on different objects or regions in the scene.

### 3.3 PRE-TRAINING STAGE

Since SCR-CCR performs recognize characters based on object-centric representations, we train an encoder, ChSA, and decoder to parse slots from input images in the pre-training stage. For an input image, the encoder extracts its visual features; the ChSA parser aggregates features that belong to the same object based on visual clues to form slots; the decoder reconstructs the features of slots and composes them into a complete feature map. The training objective of SCR-CCR is to reconstruct the DenseRAN features, allowing the ChSA to output slots that can reflect meaningful components of the character.

**Image Encoding.** The encoder is responsible for downsampling the input character image to extract visual features. Since the slot parser needs to assign each feature to one slot, the number of features is an important factor influencing the computational efficiency of SCR-CCR. On the other hand, the input character image may contain details that are not critical for recognition (*e.g.*, stroke thickness and color). By downsampling input images through the encoder, we can control the number of features and obtain more representative features for recognition. The encoder is randomly initialized and trained from scratch. Assuming that the input images $\boldsymbol{X}$, the process of image encoding is:

$$\boldsymbol{F} = \text{Encoder}(\text{Scale}(\boldsymbol{X})). \quad (5)$$

SCR-CCR scales the shape of $\boldsymbol{X}$ to $80 \times 80$ and outputs a $40 \times 40$ feature map $\boldsymbol{F}$.

**Slot Parsing.** ChSA extracts slots from $\boldsymbol{F}$, which represents different components that make up the complete character. ChSA is built upon the Slot Attention mechanism, where the features in $\boldsymbol{F}$ are iteratively assigned to different slots through update aggregation. Most character recognition datasets provide additional auxiliary information besides character images and the corresponding

categories. For example, Ideographic Description Sequence (IDS) provides a human-defined structure hierarchy and radical-level decomposition of characters. During the process of slot update, ChSA leverages auxiliary information to guide the learning of slots for more accurate inference results. Since SCR-CCR parses slots automatically, which are not designed to align with human-defined radicals, ChSA does not use meta information of radicals and only calculates the number of radicals of each training sample. As Figure 2 shows, the number of radicals is used as the number of slots in training empirically. Assuming that ChSA has $K$ slots, and the training sample contains $m$ radicals according to the auxiliary information, we introduce a $K$-dimensional indicator vector $\boldsymbol{v}$ to indicate the availability of slots, where $v_i = 1$ when $1 < i < m$, and $v_i = 0$ otherwise. ChSA changes Equation 2 to calculate the attention logits using the indicator vector:

$$\boldsymbol{A}_{i,j} = \frac{v_i \cdot e^{\boldsymbol{\Phi}_{i,j}}}{\sum_{l=1}^{K} v_l \cdot e^{\boldsymbol{\Phi}_{l,j}}}, \tag{6}$$

where $\boldsymbol{v}$ ensures that only $m$ of the $K$ slots are assignable. Controlling the number of slots encourages ChSA to learn interpretable components of characters, rather than decomposing the input into more fragmented parts.

**Feature Decoding.** The decoder is responsible for reconstructing features from the parsed slots $\boldsymbol{S}$. Most OCR models are typically trained with discriminative losses, for example, calculating cross-entropy loss on the output of a classification head. However, supervising the learning of slots with discriminative losses is hard for ChSA. The order of the parsed slots is not always consistent due to the random initialization strategy, making it difficult to determine the real category label for each slot. Besides, pre-defined radicals (*e.g.*, IDS) may not completely match the components learned by ChSA. ChSA follows the training strategy of most object-centric representation methods, *i.e.*, introducing a decoder to reconstruct the slots back into the image or features, and updating parameters through reconstruction loss. The decoder of ChSA chooses to reconstruct features because we expect the slots to contain high-level information such as component categories, rather than those used for reconstructing pixels of images (*e.g.*, stroke thickness). The decoding process is:

$$\boldsymbol{\Lambda}^k, \boldsymbol{O}^k = \text{Decoder}(\boldsymbol{s}_k), \quad k = 1, \cdots, K,$$

$$\boldsymbol{R} = \sum_{k=1}^{K} \boldsymbol{M}^k \odot \boldsymbol{O}^k, \quad \text{where } \boldsymbol{M}_{i,j}^k = \frac{v_k \cdot e^{\boldsymbol{\Lambda}_{i,j}^k}}{\sum_{l=1}^{K} v_l \cdot e^{\boldsymbol{\Lambda}_{i,j}^l}}. \tag{7}$$

$\boldsymbol{O}^k$ is the features of the $k$th component, $\boldsymbol{M}^k$ is a mask indicating the position of the $k$th component, and $\boldsymbol{R}$ is the reconstructed features. With the decoder, SCR-CCR can be trained by minimizing the distance between the reconstructed features and the features of DenseRAN. The training loss is

$$\mathcal{L} = \frac{1}{HW} \sum_{i=1}^{H} \sum_{j=1}^{W} \left\| \boldsymbol{R}_{i,j} - \bar{\boldsymbol{F}}_{i,j} \right\|_2^2, \quad \text{where } \bar{\boldsymbol{F}}_{i,j} = \frac{\boldsymbol{F}_{i,j} - \mathbb{E}[\boldsymbol{F}_{i,j}]}{\text{Var}[\boldsymbol{F}_{i,j}]}. \tag{8}$$

$\bar{\boldsymbol{F}}$ represents the standardized DenseRAN features. Since DenseRAN is trained by predicting IDS, the visual features extracted by the encoding module typically retain information related to the character structure (*e.g.*, its layout and components) while ignoring details that have less contribution to recognition. SCR-CCR leverages the feature extraction module of the pre-trained DenseRAN as the teacher encoder and fixes its parameters during the entire training procedure.

### 3.4 Slot-matching Stage

SCR-CCR uses the pre-trained encoder and ChSA to extract test slots $\hat{\boldsymbol{S}}$ from the test image $\hat{\boldsymbol{X}}$. A straightforward idea is that the distance between slots can reflect the similarity between the images. But as Figure 3 displays, even if we input the same character image, ChSA may output slots with different orders due to the randomness of slot initialization. In this case, although the input images belong to the same category, the distance between their slots can be quite large. To solve this problem, SCR-CCR uses a fixed set of vectors sampled from standard Gaussian to initialize slots. We find that the region focused by each slot is related to its initial state. If all input images share the same initial states of slots, the acquired components will tend to have the same order. SCR-CCR estimates the similarity of two images by calculating the distance (*e.g.*, L2 distance) between the ordered slots.

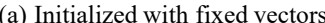

|  | Image | Slot#1 | Slot#2 | Slot#3 | Slot#4 | Slot#5 |
| --- | --- | --- | --- | --- | --- | --- |

(a) Initialized with fixed vectors  (b) Initialized randomly

Figure 3: **A comparison of slot initialization.** (a) shows the slots parsed from the fixed initial states. Slots in (b) follow the original random initialization strategy of Slot Attention.

SCR-CCR requires datasets to provide $N_e$ example images for each character in the charset $\mathcal{C}$ to illustrate different forms of the character. SCR-CCR uses the averaged example slots to represent the character $c$:

$$\bar{\boldsymbol{S}}^c = \frac{1}{N_e} \sum_{i=1}^{N_e} \boldsymbol{S}^{c,i}, \tag{9}$$

where $\left\{\boldsymbol{S}^{c,1}, \ldots, \boldsymbol{S}^{c,N_e}\right\}$ are the slots of $N_e$ examples. SCR-CCR calculates distance between $\hat{\boldsymbol{S}}$ and the averaged example slots of all characters in $\mathcal{C}$, finding the one with the smallest distance as the recognition result:

$$\hat{y} = \underset{c \in \mathcal{C}}{\arg \min} \left\| \hat{\boldsymbol{S}} - \bar{\boldsymbol{S}}^c \right\|_2^2. \tag{10}$$

## 4 EXPERIMENTS

In this section, we first introduce the experimental settings, including data construction and training details. Then, we show some results of conducted experiments (additional experimental results are shown in Appendix A.3). Finally, we conduct evaluation on Japanese and Korean characters to validate the effectiveness of SCR-CCR.

### 4.1 EXPERIMENTAL SETTINGS

**Dataset Construction.** In this paper, we mainly conduct experiments on two datasets: HWDB1.0-1.1 (Liu et al., 2013) and Printed artistic characters (Chen et al., 2021). HWDB1.0-1.1 (Liu et al., 2013) contains 2,678,424 handwritten Chinese character images with 3,881 classes, which is collected from 720 writers and covers 3,755 commonly-used Level-1 Chinese characters. Printed artistic characters (Chen et al., 2021) are generated in 105 font files and contains 394,275 samples for 3,755 Level-1 Chinese characters. Some examples of each dataset are shown in Appendix A.1. We follow (Chen et al., 2021) to construct the corresponding datasets for the character zero-shot and radical zero-shot settings. For the character zero-shot settings, we collect samples with labels falling in the first $m$ classes as the training set and the last $k$ classes as the test set. For the handwritten character dataset HWDB and printed artistic character dataset, $m$ ranges in {500, 1000, 1500, 2000, 2755} and $k$ is set to 1000. For the radical zero-shot settings, we first calculate the frequency of each radical in the lexicon. Then the samples of characters that have one or more radicals appearing less than $n$ times are collected as the test set, otherwise, collected as the training set, where $n$ ranges in {10, 20, 30, 40, 50} in the radical zero-shot settings.

**Training Details.** SCR-CCR is trained using the Adam optimizer (Kingma & Ba, 2014) where the momentums $\beta_1$ and $\beta_2$ are set to 0.9 and 0.99. For the encoder and ChSA, we increase the learning rate from 0 to $10^{-4}$ in the first 30K steps and then halve the learning rate every 250K steps. For the decoder, we increase the learning rate from 0 to $3 \times 10^{-4}$ in the first 30K steps and then halve the learning rate every 250K steps. The training batch size is 32, and the input image of SCR-CCR will be scaled to $80 \times 80$. We set the maximum number of slots as $K = 3$ in the slot-matching stage.

Table 1: **Accuracy (%) of Chinese character recognition on the character zero-shot setting.** The proposed SCR-CCR outperforms the previous methods on handwritten and printed character datasets and demonstrates outperforming recognition ability with a limited training charset (with only 500 training characters).

| Datasets | HWDB | | | | | Printed | | | | |
|---|---|---|---|---|---|---|---|---|---|---|
| | 500 | 1000 | 1500 | 2000 | 2755 | 500 | 1000 | 1500 | 2000 | 2755 |
| DenseRAN | 1.70 | 8.44 | 14.71 | 19.51 | 30.68 | 0.20 | 2.26 | 7.89 | 10.86 | 24.80 |
| HDE | 4.90 | 12.77 | 19.25 | 25.13 | 33.49 | 7.48 | 21.13 | 31.75 | 40.43 | 51.41 |
| SD | 5.60 | 13.85 | 22.88 | 25.73 | 37.91 | 7.03 | 26.22 | 48.42 | 54.86 | 65.44 |
| CUE | 7.43 | 15.75 | 24.01 | 27.04 | 40.55 | - | - | - | - | - |
| CCR-CLIP | 21.79 | 42.99 | 55.86 | 62.99 | 72.98 | 23.67 | 47.57 | 60.72 | 67.34 | 76.44 |
| Ours | **84.60** | **83.74** | **82.58** | **80.23** | **79.23** | **81.20** | **81.68** | **81.16** | **79.70** | **81.02** |

Table 2: **Accuracy (%) of Chinese character recognition on the radical zero-shot setting.** Since SCR-CCR does not rely on human-defined radical or stroke sequences for supervision, it can illustrate satisfying performance when meeting zero-shot radicals.

| Datasets | HWDB | | | | | Printed | | | | |
|---|---|---|---|---|---|---|---|---|---|---|
| | 50 | 40 | 30 | 20 | 10 | 50 | 40 | 30 | 20 | 10 |
| DenseRAN | 0.21 | 0.29 | 0.25 | 0.42 | 0.69 | 0.07 | 0.16 | 0.25 | 0.78 | 1.15 |
| HDE | 3.26 | 4.29 | 6.33 | 7.64 | 9.33 | 4.85 | 6.27 | 10.02 | 12.75 | 15.25 |
| SD | 5.28 | 6.87 | 9.02 | 14.67 | 15.83 | 11.66 | 17.23 | 20.62 | 31.10 | 35.81 |
| CCR-CLIP | 11.15 | 13.85 | 16.01 | 16.76 | 15.96 | 11.89 | 14.64 | 17.70 | 22.03 | 21.27 |
| Ours | **79.93** | **77.90** | **81.03** | **83.87** | **81.30** | **74.11** | **76.38** | **76.63** | **79.98** | **81.35** |

## 4.2 RESULTS ON CHINESE CHARACTER RECOGNITION

Two radical-based methods (Wang et al., 2018; Cao et al., 2020), one stroke-based method (Chen et al., 2021) and one matching-based method (Yu et al., 2023) are selected as the comparison methods in zero-shot settings. For fair comparison, some few-shot CCR models (Li et al., 2020), introducing the additional template samples at the training stage, are not considered. Moreover, since the character accuracy of character-based methods is almost zero in zero-shot settings, these methods are also not used for comparison.

**Character Zero-Shot Setting.** We first validate the effectiveness of the proposed SCR-CCR on the character zero-shot setting. As shown in Table 1, regardless of the handwritten or printed character dataset, the proposed SCR-CCR outperforms previous methods by a clear margin. For instance, in the 500 HWDB character zero-shot setting, the proposed method achieves a performance improvement of 62.81% compared with the previous SOTA method CCR-CLIP. However, we observe an interesting phenomenon that as the size of the training set increases, the performance of our model actually decreases to some extent. One possible reason is that although the training set includes more characters, the number of samples for each character remains unchanged. However, the proposed method relies on the differences between characters to learn compositional representations, which requires an increase in the number of samples for each character as the number of characters increases. For the interpretability of performance improvement, we have visualized some intermediate results of our method in Figure 4. More discussions are shown in Section 4.3.

**Radical Zero-Shot Setting.** Following the previous method (Chen et al., 2021), we have also conducted corresponding experiments in the radical zero-shot setting. The experimental results shown in Table 2 indicate that the proposed method achieves the best performance across all sub-settings with an average improvement of 63.12% in accuracy compared to the previous SOTA method CCR-CLIP (Yu et al., 2023). Since our method does not introduce manually defined radical or stroke sequences for supervision, the proposed SCR-CCR can still achieve satisfying performance in the case of radical zero-shot scenarios.

| Image | Slot #1 | Slot #2 | Slot #3 | Slot #4 | Slot #5 | | Image | Slot #1 | Slot #2 | Slot #3 | Slot #4 | Slot #5 | Slot #6 |

(a) Slots learned from Printed          (b) Slots learned from HWDB

Figure 4: **Visualization of the learned slots.** Although we introduce no fine-grained supervisions defined by humans, *e.g.*, radical and stroke sequences, the proposed SCR-CCR can still perceive different components with slots.

### 4.3 VISUALIZATION OF SLOTS

Previous radical-based or stroke-based methods rely on human-defined representations, such as radical or stroke sequences. Radical-based methods suffer from inconsistent decomposition across different characters, which requires the model to extract different features from the same visual characteristics, thereby hindering performance. In addition, stroke-based methods require perceiving fine-grained stroke information, which is challenging for Chinese characters with complex structures. In this section, we attempt to visualize the compositional representations learned by the proposed method. As shown in Figure 4, we visualize the regions attended by different slots for both printed and handwritten character samples. The visualization results reveal that each slot focuses on distinct and independent components of the characters. It is satisfying to note that despite the absence of any fine-grained supervision information (such as radicals or strokes), different slots can still effectively distinguish different character regions. Therefore, the compositional representations learned in an unsupervised manner from the training set characters can possess stronger generalization capabilities, thus being more robust to zero-shot characters.

### 4.4 DIFFERENT NUMBER OF SLOTS AND EXAMPLES

In experiments, we observe that there are two factors that affect the performance of SCR-CCR in inference: the number of slots and that of template character images used for matching. As shown in Figure 5, we have evaluated the performance of SCR-CCR with different slot and template character image quantities in the character zero-shot settings. On both HWDB and Printed characters, SCR-CCR exhibits better performance when the number of slots is set to 2 or 3. This also conforms to the way that native Chinese speakers recognize Chinese characters, *i.e.*, they tend to only focus on the patterns of left-right or upper-lower radicals of the entire character. In addition, we find that the more template characters used in inference, the better the performance of SCR-CCR. Therefore, we use 10 character images for matching in the final experiments. It should be noted that since the features of character images used for matching can be extracted in advance, there will be no additional inference time when more template character images are used. More experimental results in the radical zero-shot setting are displayed in Appendix A.3.2.

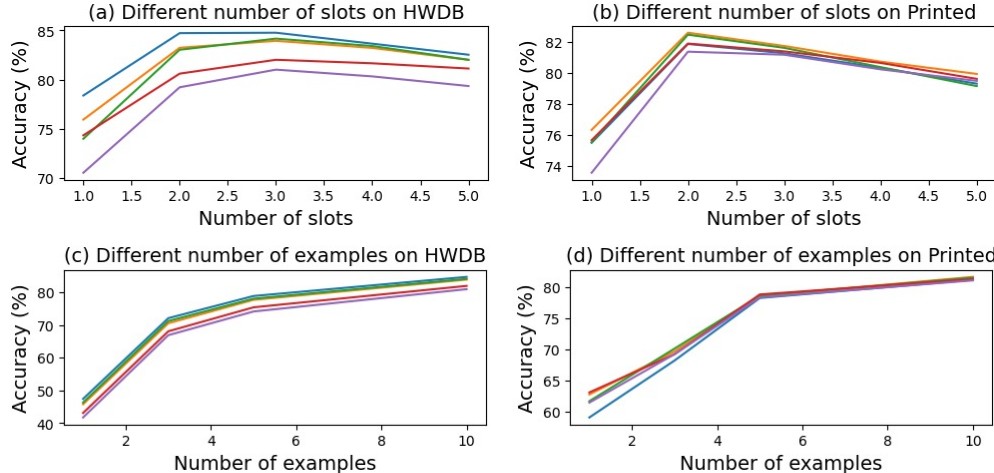

Figure 5: **Accuracy on the different number of slots and examples.** (a) and (b) illustrate how the number of slots influences the accuracy of slot-matching. (c) and (d) show the impact of the number of examples on the experimental results.

Table 3: **Accuracy (%) of recognizing Japanese and Korean characters.** The proposed SCR-CCR can achieve satisfying performance in recognizing Japanese and Korean characters with a pre-trained model trained only on Chinese characters.

| Datasets | Japanese | Korean |
|----------|----------|--------|
| Ours     | 89.46    | 62.13  |

## 4.5 Zero-shot Japanese and Korean Character Recognition

Most Chinese zero-shot character recognition methods can only achieve limited generalization on unseen Chinese characters and cannot further generalize to other languages. To demonstrate the effectiveness of their method on character recognition in other languages, Chen et al. (2021) defined stroke sequences for Korean characters using their stroke-decomposition method and achieved satisfying performance in Korean character recognition. However, this method still requires collecting Korean character images for training. Unlike the evaluation of generalization in previous methods, we directly perform testing on Japanese and Korean characters without any training or fine-tuning on data of these languages. The experimental results shown in Table 3 indicate that, despite not being trained on any Japanese or Korean character datasets, the proposed SCR-CCR achieves an accuracy of 89.46% and 62.13% on Japanese and Korean characters, respectively.

## 5 Conclusion

In this paper, we introduce the **S**elf-learning **C**ompositional **R**epresentation method for zero-shot **C**hinese **C**haracter **R**ecognition (**SCR-CCR**) to address challenges in Chinese character recognition, particularly zero-shot recognition. SCR-CCR offers a unique solution by autonomously learning compositional components from the data, distinct from traditional radical or stroke-based approaches. By following a pretraining-inference paradigm and leveraging Character Slot Attention, SCR-CCR excels in extracting relevant components for recognition. The experimental results demonstrate that SCR-CCR outperforms previous methods in most scenarios of character and radical zero-shot settings. Particularly, visualization experiments reveal that the components learned by SCR-CCR reflect the structure of characters in an interpretable manner and can be applied to recognize Japanese and Korean characters. In essence, SCR-CCR not only advances the field of Chinese character recognition but also offers insights into broader applications across languages.

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

## A APPENDIX

### A.1 EXAMPLES OF ADOPTED DATASETS

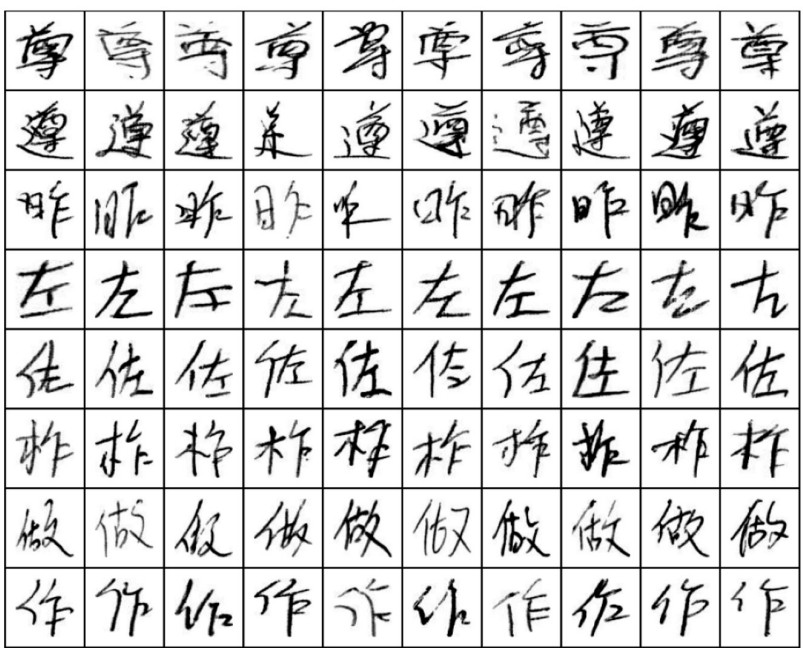

(a) Examples of handwritten characters

(b) Examples of Printed characters

Figure 6: Visualization of the adopted datasets (a) HWDB and (b) Printed.

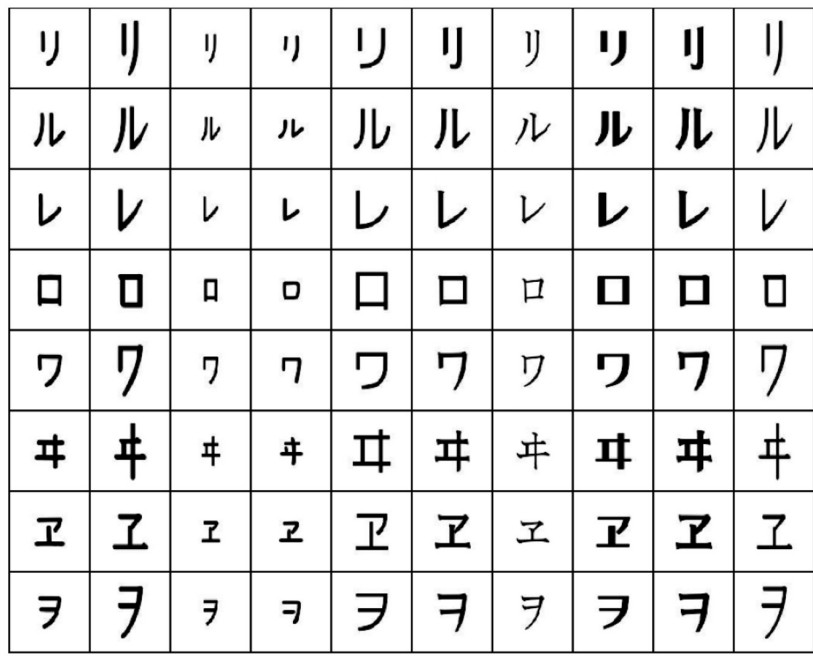

(a) Examples of Korean characters

(b) Examples of Japanese characters

Figure 7: Visualization of the additional (a) Korean and (b) Japanese test characters.

## A.2 DETAILS OF SCR-CCR

This section describes the architectures of learnable networks in SCR-CCR, including the encoder and decoder. The architecture of ChSA follows the original design of Slot Attention (Locatello et al., 2020).

- **Encoder**:
    - $5 \times 5$ Conv, stride 2, padding 2, 192, ReLU
    - [ $5 \times 5$ Conv, stride 1, padding 2, 192, ReLU ] $\times$ 2
    - $5 \times 5$ Conv, stride 1, padding 2, 192
    - Cartesian Positional Embedding, 192, LayerNorm
    - Fully Connected, 192 ReLU
    - Fully Connected, 192
- **Decoder**:
    - Fully Connected, 192 ReLU
    - Learnable 2D Positional Embedding, 192
    - [ Fully Connected, 1024 ReLU ] $\times$ 2
    - [ Fully Connected, 1024 ] $\times$ 2

## A.3 ADDITIONAL EXPERIMENTAL RESULTS

### A.3.1 CLUSTERING OF CHARACTERS

To further validate the effectiveness of the representations learned by SCR-CCR, we cluster the whole features of slots and visualize the clustering results in Figure 8. The results demonstrate that SCR-CCR can effectively distinguish different characters in the feature space, whether on handwritten or printed Chinese characters. Interestingly, compared to handwritten characters, the features of printed characters exhibit more ambiguity in the feature space. While printed characters are typically easier to recognize (*i.e.*, the features of printed characters are more distinguishable), the examples of adopted datasets shown in Appendix A.1 indicate that the diversity of printed characters is no less than that of handwritten characters.

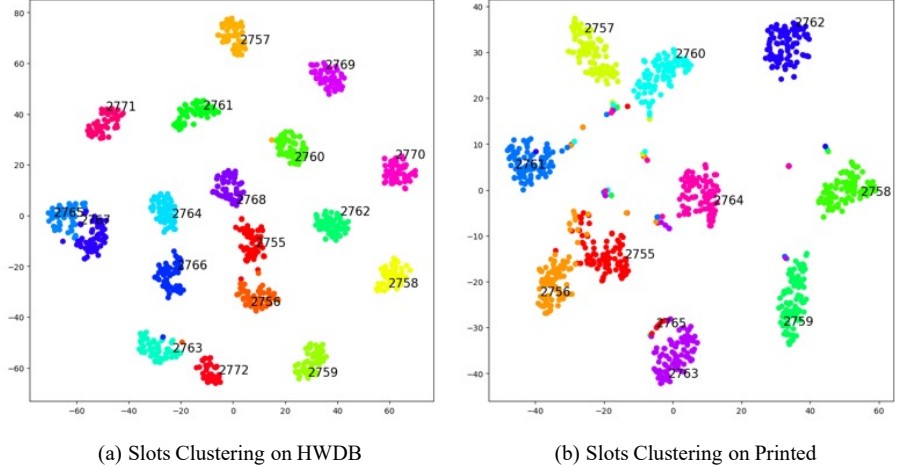

(a) Slots Clustering on HWDB       (b) Slots Clustering on Printed

Figure 8: **Visualization of the slot clustering.** The slot clustering results demonstrate that the learned compositional representations can be well distinguished in the feature space.

### A.3.2 DIFFERENT NUMBER OF SLOTS AND EXAMPLES

We also explore the impact of different numbers of slots and template character images on model performance in the radical zero-shot setting. The experiment indicates that when the number of slots

is set to 2 3 and the number of template character images is set to 10, SCR-CCR can achieve the best performance, which is consistent with the conclusion in Section 4.4.

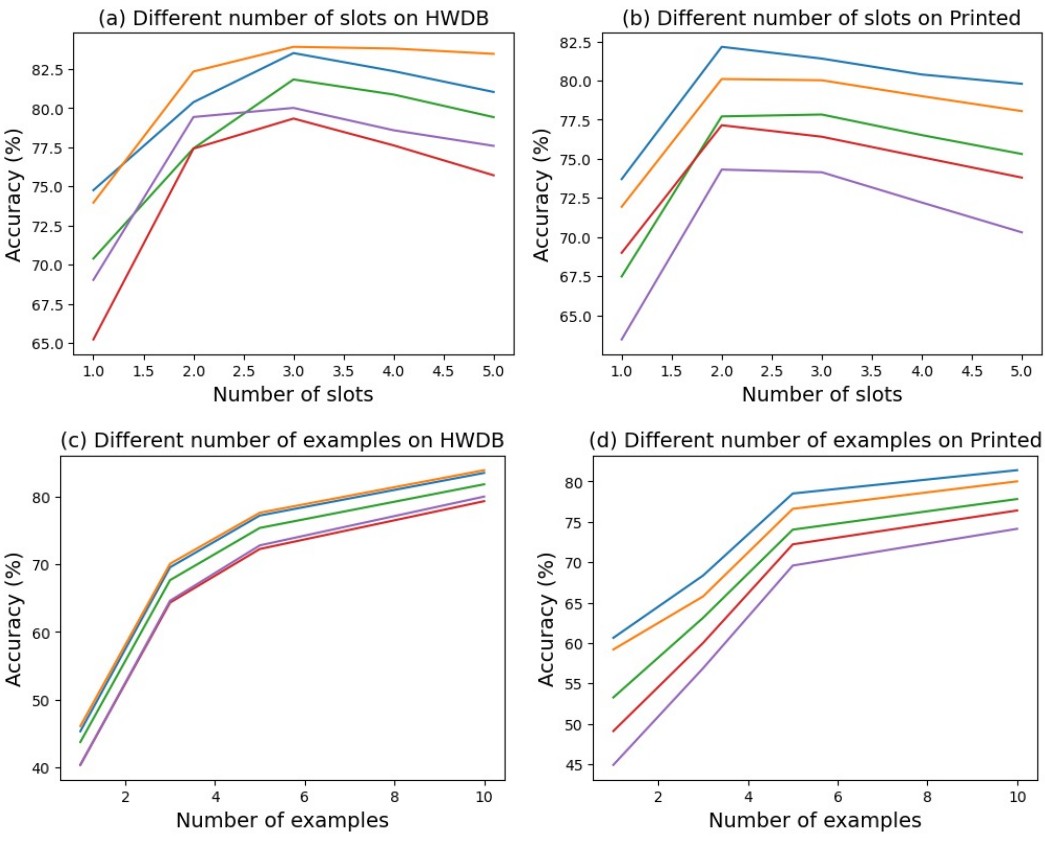

Figure 9: **Accuracy on the different number of slots and examples (radical zero-shot setting).** (a) and (b) illustrate how the number of slots influences the accuracy of slot-matching. (c) and (d) show the impact of the number of examples on the experimental results.

### A.3.3 CANDIDATES IN SLOT-MATCHING

As shown in Figures 10-12, we list the candidates that the model considers most similar to the input image during the slot-matching process. Overall, SCR-CCR tends to confuse characters with the same radicals or structures. This is particularly common in Chinese and Korean character recognition since they have similar hierarchical structures. In Japanese, confusion typically occurs in the symbols located in the upper right corner. Since Japanese characters have relatively simpler structures, component-level confusion occurs less frequently.

### A.3.4 ZERO-SHOT JAPANESE AND KOREAN CHARACTER RECOGNITION

Although SCR-CCR is trained on pure Chinese data, we still attempted to visualize the slots parsed from Japanese and Korean data in this experiment. The visualization results in Figure 13 show that, SCR-CCR can still discover some meaningful components in Korean data. This might be due to the model having learned layout-related knowledge from the Chinese data, enabling it to parse unseen components from non-Chinese characters. In contrast, Japanese characters have simpler structures, and the model tends to recognize the entire Japanese character as a single component.

挖：[挖],饱,惨,袍,掺,狡,浚,抱,修,讫,竣,馆,拽,胞,骏,按,榨,穆,梭,狼,饺,俊,鞍,拔,挨,腔,独,炮,搜,馋,鲍,谬,脓,悔,娘,胶,彼,谊,控,拔
吸：[吸],圾,极,服,叹,报,及,汲,版,股,叨,板,吗,顺,暇,贩,饭,帆,陨,限,恢,设,投,胺,收,殴,恨,眠,帧,坡,般,眼,役,怀,撅,峡,段
虞：[虞],侯,虐,晨,褒,詹,良,矣,侵,集,度,复,虎,屋,层,拿,震,冥,窄,窟,袭,展,衰,窟,候,食,岸,袋,崖,贫,寡,停,宴,像,赏,阜,龚,皋,蔑,危
造：[造],选,送,迭,遣,连,遗,筐,注,佳,崔,挂,拦,进,徒,进,违,搓,蓬,道,谱,迷,雀,借,蓬,佐,逞,桂,遂,指,性,递,烂,崖,佬,催,逆,住,哇,谨
宣：[宣],宜,宦,宝,富,室,宫,宫,莹,壹,皇,宴,盲,昼,窟,屋,鱼,毫,冥,直,拿,言,享,童,筐,置,窒,冒,堂,彦,章,豆,复,营,蔓,萤,星,量,邑
扎：[扎],礼,札,孔,轧,乱,化,升,扑,北,抖,比,什,补,孙,仇,扒,外,执,牡,朴,乳,儿,杜,让,仕,认,社,仆,斗,壮,计,巩,仙,仇,扯
债：值,信,猿,侯,惰,[债],侍,请,借,侵,傍,使,搞,伎,候,筐,填,传,住,循,伟,伴,情,撞,赁,蔼,倚,停,偕,懂,傅,搓,掉,辖,崔,慎,佬,瘴,悼,噎
氧：[氧],象,氦,氢,氰,秉,氢,龟,氖,氮,享,争,免,弟,阜,重,氟,泵,章,氖,拿,单,身,氯,蔓,竟,牵,衷,系,色,辜,勇,袁,束,直,夷,毫,事,膏,亥
兄：[兄],见,冗,兄,另,足,贝,呆,晃,兑,吊,尺,灵,觅,元,尼,央,员,朵,吴,异,完,况,民,买,昆,吕,早,几,已,昂,采,灭,品,界,免,己,泉,显
咬：[咬],饺,狡,校,孩,咳,该,胶,陕,眩,该,恢,依,疚,弦,舷,峡,较,侠,饺,咬,绞,狭,坟,腋,胶,骇,族,陵,脓,挟,疾,吨,枝,破,咙,按,陀
毋：丹,尹,月,勺,习,刀,刀,匀,勿,冉,勿,力,刁,夕,为,勺,冈,坍,母,目,勾,闩,日,用,舟,日,伊,甲,[毋],司,闭,内,身,伪,罚,凡,肉,旬,匈,田
循：[循],惦,惰,框,猛,候,恬,糖,任,隔,婚,桩,据,陌,裤,佰,低,惟,谚,侄,侯,柜,椎,佳,俯,隋,恒,幅,桓,作,嘘,悟,振,猴,桂,悔,指,福,括,桅
影：[影],彭,彰,彩,彪,乾,乾,彤,杉,骏,敦,敦,歇,欺,敲,数,掀,款,敏,形,鼓,勤,勤,敛,敬,丝,致,勋,勤,鹤,韵,勃,孰,澎,斯,叙,鄙,酞
枝：技,[枝],伎,扶,吱,抹,妓,佑,找,仗,狡,校,拔,怯,获,拄,袜,休,拢,佐,枯,歧,传,狡,坟,估,妹,块,伐,挟,依,肢,坡,拣,核,抉,疾,棱,猿,该
卸：一,丫,凹,以,扩,加,口,扑,扣,叭,凶,扎,矿,扒,印,叫,讣,朴,仙,四,山,比,卜,札,外,小,知,办,曲,拟,如,认,协,少,似,汕,忆,仇,孙,功
围：[围],囤,闺,国,圆,图,圈,囤,圈,困,闯,固,阁,周,团,囚,阁,阎,阀,同,阔,闻,闽,闹,阑,阐,凰,囱,肉,回,闯,甩,用,闸,冈,阅,司,闭
植：桩,殖,[植],拍,值,桓,柏,柜,拄,柱,佑,枯,掉,拒,捕,狂,椎,枝,堆,拍,性,桂,恒,佳,推,佐,柿,伯,挂,埔,拣,垃,拉,栋,作,住,施,佔,梅,任
诈：作,[诈],柞,竹,咋,件,昨,炸,非,伤,许,休,仿,价,忙,佐,护,狞,徘,体,乍,炉,饰,代,犹,炉,铲,状,扶,仔,什,诽,仟,仲,筛,怀,协,饼,拼,钵
走：[走],圭,志,击,主,未,末,玉,表,去,老,毛,赤,丢,夫,盂,壶,承,左,王,忘,丰,吉,壬,羌,庄,衣,屯,歪,孟,朱,茫,夹,在,违,禾,来,无,定,羌
委：蚕,歪,至,[委],季,丢,秀,香,医,黍,玉,妄,垂,萎,盂,番,吾,紊,重,奎,玄,圭,蛋,套,奢,畜,羹,亥,吞,奏,秦,蠢,垄,壶,豪,变,否,卖,素,蛮
啸：佣,阔,闹,[啸],啤,伸,佛,倔,嘛,佩,隅,碉,阀,谰,窟,调,律,肉,唱,阁,闷,闵,偶,闲,使,庸,侗,阔,凋,琅,绸,坤,曳,呢,寝,埔
赵：[赵],凶,赴,凹,汹,缸,蚁,趾,讥,议,幽,逊,砒,弘,酗,处,汕,比,达,灿,卧,毗,认,巡,毯,私,越,让,矿,斌,扯,挞,改,挝,以,耻,双,孙,忆,肛
芽：芋,[芽],井,羊,芦,车,丰,苛,卉,苯,年,芹,手,茅,耳,弃,苏,茫,苹,牛,苟,东,朱,井,节,声,赤,抒,茁,萧,芍,牙,杀,寺,奇,示,未,未,萍,半
域：城,[域],找,球,拭,试,械,诚,协,状,拢,绒,戏,诚,伏,忧,斌,休,扶,拔,饿,抹,娥,铱,绒,扰,戎,饺,战,减,袜,犹,狱,碱,伐,优,铱,妹,俄,吠
州：叫,[州],川,丫,竹,训,们,外,似,以,升,州,纠,付,刊,从,判,村,升,朴,仰,卜,刑,料,门,削,抖,印,讨,扩,旷,似,矿,列,叶
猪：[猪],稍,祥,猎,捐,梢,捎,揖,伪,猬,样,悄,俏,拍,猫,措,拷,辑,捏,佯,惰,摧,猖,蹄,指,梯,佬,拼,储,诸,拦,谚,捞,柏,借,消,栏,狰,硝,榨
葬：[葬],莽,蒜,菇,芽,茫,狄,薛,药,靠,茅,萧,荐,苯,彝,蔑,茂,范,森,藉,兼,幸,葫,燕,著,素,年,蓉,茄,茬,葵,雍,莱,菲,苹,萎,苇,泰,蔬
喻：偷,愉,榆,[喻],渝,输,喻,喻,嗡,余,俞,伤,悔,确,响,价,偷,你,筛,惊,俞,翁,伯,痈,晌,仿,怖,痢,狗,愉,煽,晦,侗,哺,殉,诲,馅
著：着,养,表,羌,菩,卷,巷,青,考,芜,[著],希,差,茅,昔,昔,券,袁,关,蓄,羊,孝,苇,善,老,姜,肯,春,芽,告,裹,丢,看,萎,尧,袭,幕,言,韦,毒
淫：[淫],谣,湿,汪,浑,逞,浮,瑶,遥,滔,澡,溪,诬,深,涅,滓,混,噪,垣,涩,泽,诏,谨,捉,逼,评,馁,泥,强,俘,译,摇,误,军,淬,涯,保,促,逗,远

Figure 10: **Candidates in slot-matching of Chinese characters.** The degree of matching decreases from left to right. The matching targets are indicated in square brackets.

タ：[タ],ク,ヌ,ケ,ダ,メ,グ,つ,ろ,り,ゲ,マ,カ,め,ス,ウ,プ,フ,づ,ズ,ら,ち,や,ブ,の,ぐ,ヲ,ヤ,セ,く,ぬ,ゾ,コ,う,バ,ゆ,ぺ,ヱ,よ,ン
ダ：[ダ],グ,タ,ゲ,ぢ,ク,ヴ,ガ,ズ,づ,ケ,ブ,そ,が,ぜ,バ,メ,だ,プ,バ,ザ,な,ゴ,デ,ヌ,べ,ず,ぜ,ど,ざ,か,べ,べ,ぺ,お,り,め,ぶ,ヂ,ぐ
チ：[チ],テ,モ,キ,エ,ヲ,モ,フ,エ,ろ,オ,す,て,ア,ナ,チ,ユ,コ,こ,ら,ま,ケ,ニ,で,きる,チ,え,ち,ネ,さ,ス,丁,マ,プ,デ
ヂ：[ヂ],デ,プ,プ,ず,シ,チ,ナ,ギ,づ,す,オ,ゴ,み,か,び,ズ,ン,の,ゾ,ご,ポ,ぴ,ひ,ザ,つ,ケ,ヲ,ゲ,ナ,フ,ヤ,そ,げ,ヴ,ア,サ,イ,ソ,ガ
ツ：[ツ],ソ,ン,ゾ,シ,ヅ,り,ジ,リ,メ,の,つ,レ,じ,プ,サ,ひ,プ,ブ,ル,ザ,ゆ,め,ヴ,い,グ,ク,ゲ,び,や,け,ウ,ケ,し,げ,ビ,フ,ぴ,ワ,ベ
ヅ：[ヅ],ゾ,ジ,ツ,ヴ,プ,ブ,シ,づ,ウ,ン,ゲ,ソ,ク,ケ,つ,ぐ,プ,ザ,ワ,り,ダ,ら,げ,ろ,ダ,ヂ,ゴ,ゆ,デ,サ,め,り,ろ,バ,ヌ
テ：[テ],ラ,チ,う,フ,ヲ,エ,ア,ら,ろ,こ,て,そ,モ,え,ヱ,デ,コ,ケ,ニ,キ,マ,了,ス,ネ,丁,で,キ,す,ヨ,こ,ちる,ブ,ふ,さ,ナ,ヌ,オ,き
デ：[デ],ブ,プ,ゴ,ヂ,づ,ず,ギ,ヴ,ざ,ゲ,ご,ズ,ボ,ポ,す,ゾ,ナ,テ,そ,ぶ,ジ,ぎ,バ,ガ,オ,ザ,チ,グ,ヰ,シ,さ,ぢ,ケ,ぶ,が,ど,ぜ,バ,ト
ト：[ト],ト,へ,へ,ド,ハ,く,い,し,レ,よ,じ,ぐ,り,ヒ,ひ,メ,ン,个,ム,ナ,ビ,べ,べ,と,か,ホ,バ,ハ,イ,ん,ケ,ふ,リ,ル,ソ,ペ,ヤ,べ,卜
ド：[ド],ト,く,ぐ,ト,ハ,よ,じ,バ,り,ビ,ペ,ペ,ベ,い,へ,バ,へ,ど,ビ,メ,リ,か,に,ヒ,ケ,ふ,ひ,ヤ,イ,ポ,や,の,ボ,と,ズ,ヌ,ホ,マ
ナ：[ナ],す,オ,サ,ブ,ン,プ,ケ,よ,ホ,ゾ,ず,ソ,づ,ヰ,ゴ,ノ,メ,キ,シ,つ,オ,け,フ,と,イ,ヤ,せ,か,カ,レ,さ,コ,こ,ザ,く,ど,ク,ご,ナ
ニ：[ニ],エ,こ,ユ,ヱ,コ,ご,ニ,て,さ,ヨ,ゴ,テ,に,と,エ,マ,フ,よ,ミ,う,ス,そ,ラ,ら,ロ,ヰ,ケ,ナ,ヌ,で,モ,チ,キ,ろ,ヒ,ヲ,ふ,ち
ヌ：[ヌ],ス,マ,フ,メ,て,ろ,つ,ヱ,コ,ユ,ズ,タ,ヲ,く,こ,ワ,ア,ロ,プ,ク,う,ら,り,カ,の,み,そ,ふ,ヨ,で,よ,ブ,ケ,ベ,ウ,ご,め,イ,ヤ
ネ：[ネ],ホ,ス,ふ,こ,ヌ,タ,ヌ,ヲ,モ,で,ヨ,ヲ,ウ,ユ,そ,テ,フ,ラ,ヤ,さ,ま,ケ,ろ,て,エ,や,カ,き,チ,ヰ,こ,オ,ミ,つ,よ
ノ：[ノ],メ,ン,ソ,イ,ゾ,プ,ブ,レ,つ,フ,く,シ,ん,ナ,づ,ク,ハ,ツ,バ,ル,ム,リ,リ,し,の,ケ,ゴ,ジ,ベ,ヌ,ヅ,よ,グ,い,じ,べ,と,べ
ハ：[ハ],へ,へ,い,バ,べ,べ,ム,ベ,バ,ペ,く,ル,ハ,卜,リ,メ,リ,か,レ,ん,じ,の,ひ,ソ,ぐ,ド,レ,ン,イ,ズ,よ,卜,ビ,ど,ノ,从,ヒ,ル,ツ
バ：[バ],バ,べ,ハ,べ,ペ,ペ,い,へ,へ,く,ズ,メ,ド,ゲ,ビ,ぐ,ど,が,か,り,ボ,ル,の,よ,じ,ポ,リ,ム,ブ,ガ,イ,ヴ,ザ,グ,ん,ぴ,ひ,ひ,げ
パ：バ,[パ],ペ,ペ,べ,べ,ハ,い,ペ,べ,ズ,ビ,の,どり,が,ど,り,ぶ,が,か,ビ,ル,び,ド,ぐ,じ,ソ,リ,ポ,ブ,ひ,ガ,よ,イ,ソ,ザ,ぴ,プ,ボ,げ
ヒ：[ヒ],と,じ,ビ,ピ,し,セ,レ,に,ど,こ,せ,よ,もん,て,た,ヤ,さ,く,ル,れ,ロ,卜,キ,ナ,ひ,ご,や,ぜ,む,ン,ち,ニ,ぜ,北,い,ケ,モ,シ
ビ：[ビ],ピ,ど,ヒ,と,じ,せ,ぜ,ご,こ,に,さ,よ,セ,ざ,し,レ,ロ,べ,ベ,びく,べ,む,ゴ,ル,ゾ,ひ,ベ,ナ,ぞ,は,ん,バ,ヱ,ば,て,け,め
ピ：[ピ],ビ,ど,ヒ,と,じ,に,こ,ご,ぜ,せ,ぜ,セ,よ,さ,ロ,ヱ,ざ,く,ゾ,て,べ,べ,し,レ,ユ,ご,ん,ニ,バ,の,ン,ヤ,ば,べ,ぞ,ゆ,ナ,ル,は
フ：[フ],つ,ヲ,コ,ア,マ,ヌ,エ,ろ,ヌ,ラ,ウ,エ,ろ,ヱ,ツ,カ,ケ,ティ,イ,て,エ,ら,ら,オ,ソ,づ,ナ,了,ヤ,こ,り,刀
ブ：[ブ],プ,づ,ゴ,ゾ,フ,つ,ン,デ,ジ,ナ,す,ヲ,ソ,オ,ず,コ,シ,ゾ,ノ,ク,ヴ,メ,マ,ズ,ケ,グ,ゲ,ヌ,ユ,ア,ヱ,ウ,イ,ご,ヂ,ら,ツ,ガ,ろ
ヘ：[ヘ],へ,ハ,べ,ベ,い,ペ,べ,バ,く,ム,卜,ハ,メ,ン,バ,レ,卜,じ,り,し,ひ,ソ,ル,ぐ,ん,の,ズ,か,よ,ド,つ,ヒ,ビ,リ,从,ヌ,大,ス,ジ
ベ：ベ,[ベ],べ,べ,バ,パ,ヘ,へ,ハ,ズ,メ,い,ぐ,ど,ご,ど,ビ,ブ,ボ,よ,び,ぶ,ゲ,か,ひ,ソ,ム,ド,り,ポ,ザ,が,ビ,じ,ヴ,ピ,ぜ,ざ
ペ：べ,べ,[ペ],べ,バ,ヘ,ヘ,バ,ハ,メ,く,ズ,ビ,どい,ご,じ,よ,ぐ,ピ,ン,ヌ,づ,ひ,び,との,り,か,ぶ,こ,に,ソ,ド,ポ,ボ,つ,て,ブ,ル
ホ：[ホ],ボ,す,ポ,オ,よ,ま,キ,ヰ,オ,ナ,木,ふ,さ,ネ,寸,ず,ケ,ち,卜,チ,か,カ,不,サ,六,本,マ,ぶ,て,ギ,ヌ,ら,き,ウ,ご,こ,末,ヤ,テ
ボ：[ボ],ポ,ホ,ギ,ず,ぶ,ざ,か,バ,が,よ,デ,ズ,ま,べ,べ,ぎ,ゴ,ガ,す,お,オ,ヂ,ヴ,ブ,な,ゲ,ザ,ぶ,ぢ,ば,ヰ,そ,べ,さ,バ,ペ,づ,ナ,む
ポ：ボ,[ポ],ホ,ぶ,ず,ギ,か,が,ざ,デ,ぶ,よ,バ,ズ,ガ,ま,す,ヂ,ザ,お,ゴ,オ,べ,べ,ぎ,ヴ,ブ,ゲ,ば,バ,ベ,ヰ,な,そ,べ,ぢ,び,ご,む,ナ

Figure 11: **Candidates in slot-matching of Japanese characters.** The degree of matching decreases from left to right. The matching targets are indicated in square brackets.

큉: 킁,[큉],귕,큄,큄,큉,퀼,뀐,퀭,꿜,컁,퀀,큅,갱,뀜,핑,꿩,킬,컁,큄,뀐,겜,퓝,검,긩,뀝,퀼,킹,징,췸,침,핌,킹,킴,켕,컴,겅,뀐,정,긴
교: 고,그,[교],크,코,끄,꼬,표,뽀,므,즈,프,근,표,묘,포,균,곤,요,모,르,오,조,쯔,뜨,으,죠,군,끈,드,료,도,구,로,뇨,또,규,흐,근,굿
구: [구],규,꾸,쿠,극,국,긍,군,근,금,규,푸,곡,주,근,궁,큐,긋,곤,무,급,군,긋,꾹,쥬,굠,퓨,꾹,굼,포,뮤,공,곳,긋,끈,우,글,크,루
국: [국],곡,극,꾹,궁,금,긍,쿡,굼,긂,꺽,콕,공,쿡,퓨,곰,급,긥,구,긋,긁,긎,푹,푸,꼭,꿍,규,굴,긁,군,긋,쿠,글,곱,근,목,꿈,묵,죽,쿵
군: [군],곤,근,근,군,균,곤,꾼,긎,끈,곤,글,곤,글,푼,긍,끈,궁,꾼,긋,굴,긋,굼,곳,긎,굴,굴,긊,퓬,공,긎,코,꾼,포,꼰,푼,꾿,르,꿈
굔: [굔],귿,근,곤,귤,글,굴,곤,꿜,긎,꿏,군,긊,곤,굲,긎,긎,굲,꾼,군,균,공,긎,료,큰,긍,푠,끝,긊,긎,꿋,꿈,쿵,로,곱,론,긥,곰,쿤,긅,뭍,근
귤: 귤,굴,[귤],글,골,꿂,곤,굔,군,긊,긎,긮,쿨,글,꿀,꿀,굡,귭,긊,긋,끝,긊,귤,끝,꿇,공,곰,묠,금,곳,귬,긎,큘,긍,큠,뭍,궁,궈,쿵,곰,쿰
굶: [굶],꿂,긊,긥,긂,긥,긞,귤,긥,긂,굖,꿊,긂,굪,곪,긃,긝,공,긘,골,긥,꿉,긍,긋,쿰,곰,콩,쿵,줌,곤,꿈,꿀,곱,쿨
긂긂: 긂,[긂],꿂,굚,귆,긃,긃,귒,굒,긊,근,긂,긥,귯,글,�73,곤,긊,긚,곫,꿁,긙,풍,쁂,쿰,쿰,쁭,쁭,근
굼: [굼],긋,긍,궁,곰,긝,공,긩,굴,긥,긂,귤,굮,글,긂,굤,곫,긂,곤,근,근,긂,쿔,곰,쿵,굮,긁,긎,끔,굣,큼,쿰,콩,국,긩,긎,굡
굽: [굽],급,긋,곱,긊,�Ũ,긋,굴,급,곰,공,군,긍,글,긂,굨,긨,곩,곪,골,긂,긝,꿈,곣,군,굨,긆,긡,긗,굣,곱,큼,�99,꿀,쿵,쿰,믑,콤,긎
굿: [굿],긋,긎,긎,긎,긎,긎,콧,긋,꼿,긎,곤,근,긍,군,궁,균,긋,근,극,줏,뭇,굣,국,못,규,구,긎,풋,글,곡,긃,금,긎,풋,곤,픗,규
궁: [궁],공,굼,긍,긂,긁,긁,긁,굴,글,궐,긁,군,급,쿵,꿍,굵,골,굵,글,긍,근,긄,곫,긂,콩,긍,국,꿈,쿰,곡,굣,긂,긎,굴,글,곰,긂,풍
굦: [굦],굦,굦,긋,굦,긁,긂,꿋,콧,긎,롯,긋,뭊,군,굤,굲,긎,긂,긁,근,꿈,긋,돗,굲,곪,굲,긎,긞,굲,뭊,공,공,긂,귬,꿘,꿋
귀: [귀],귀,궤,긔,뀌,꿔,긔,겨,꿔,꿔,뀌,겨,러,건,저,거,쥐,게,져,계,커,궈,컨,긴,퍼,펴,줘,케,긴,꺼,긔,레,권,레,려,뀌,죄,건,개,제,긴
궉: [궉],긕,꿕,긕,쿼,킥,켁,긘,격,퍅,긩,픽,긕,객,젘,퍼,릭,럭,겅,궏,궏,캑,곽,�?,팩,껙,긓,렉,객,긴,궹,경,직,끽,겨,펙,팩,킹,긤,컹,긤
권: [권],긘,[권],긴,긴,긘,꿘,컨,건,꿩,관,겐,꾼,핀,꾼,긴,킨,쿤,쥔,켼,진,린,긥,견,갠,긴,길,긩,긘,긮,긘,컨,궹,겐,궘,긩,긩,간,찐,진
궐: [궐],긜,꿜,궐,긘,컬,긜,컬,겥,럴,긤,겔,컐,질,필,칩,괄,견,꿥,셀,쬔,곌,긘,펄,졀,꽬,걸,궹,월,쉴,긜,건,긩,궠,꾜,걜,쬘
궝: 궝,[궝],긣,궳,궝,겅,궐,긘,긥,꽹,긘,궹,킹,꿩,긥,긩,긥,킹,긬,궈,궹,경,정,긴,징,긩,컵,겜,긕,긩,컴,�킹,킹,징,권
궤: [궤],게,케,계,개,궈,레,제,꿔,꿰,캐,레,꿰,꿔,개,재,괘,긕,궤,쉐,래,쵀,헤,헤,대,페,데,패,긕,쾌,께,괴,에,겐,체,째,깨,뒈,꺼,갠
궹: 궹,[궹],궹,긩,겜,궝,긣,긎,깅,궝,긥,긤,궐,긩,권,꿘,켐,궐,갱,꿤,긤,껍,꾼,겔,꿘,꾕,

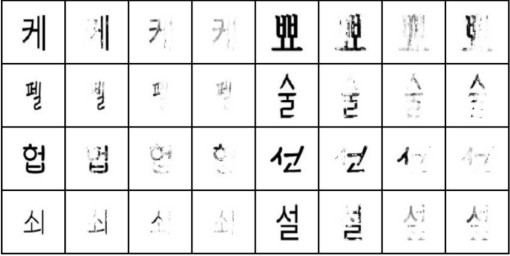

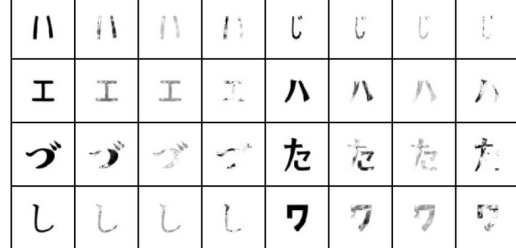

(a) Slots learned from Korean          (b) Slots learned from Japanese

Figure 13: Visualization of the slots learned from (a) Japanese and (b) Korean characters.