# OpenReview forum: "Self-learning Compositional Representations for Zero-shot Chinese Character Recognition"
_ICLR.cc/2025/Conference — ICLR 2025 Conference Withdrawn Submission_

### Official Review · Reviewer_aTUs · 2024-10-31

**Soundness:** 2
**Presentation:** 3
**Contribution:** 2
**Rating:** 5
**Confidence:** 4

**Summary:**

Summary.
This work proposes a novel zero-shot Chinese character recognition method that learns compositional representations automatically, bypassing human-defined radicals or strokes. The approach generalizes effectively to unseen characters through a Character Slot Attention (ChSA) mechanism, achieving substantial gains in zero-shot recognition across Chinese, Japanese, and Korean scripts. Experimental results highlight significant performance improvements and structural interpretability, positioning this method as a promising advancement in compositional zero-shot learning for character recognition. However, there are some key issues that the authors need to clarify in the paper.

**Strengths:**

Strengths.
1. This paper introduces an object-centric representation learning mechanism and presents a novel paradigm for zero-shot Chinese character recognition (CCR). A major strength of the method lies in its self-learning approach, which parses slot features (the model's own learned compositional understanding of Chinese characters) directly from data rather than relying on human-defined radicals or strokes. This contributes to the model's improved generalization capability.
2. The paper includes multiple zero-shot character recognition experiments demonstrating the effectiveness of the proposed method and yielding promising results.
3. Additionally, the authors provide clear expression and logical organization, effectively communicating their core contributions and the model's operational details.

**Weaknesses:**

Weaknesses.
Despite the promising experimental results, several key issues require further clarification:
1. In the slot-matching stage, SCR-CCR needs to prepare 10 example images for each character class to be used as slot information for this class to help with similarity matching. However, it significantly compromises zero-shot learning (ZSL) principles, as unseen character classes should not have accessible example images. Thus, the fairness of the comparative experiments is questionable.
2. SCR-CCR matching mechanism resembles template-based methods, relying on auxiliary example images. This approach may lack practicality in real-world settings where many uncommon character classes exist, and obtaining enough example images can be challenging. In contrast, template images are more easily synthesized by computers.
3. In Section 4.2, Character Zero-Shot Setting, the authors indicate that the proposed method depends on learning compositional representations based on character differences, which echoes the issues mentioned in the second point.
4. In the last sentence of the second paragraph of the Introduction, the authors state that "....these methods may lack the flexibility to adapt to different scenarios and have poor generalization in practical applications." To support this claim, additional zero-shot character experiments in natural scenes, such as on the CTW dataset, would be needed to demonstrate the proposed method's practicality and generalization over prior approaches.

**Questions:**

Please refer to my previous comments.

---

### Official Review · Reviewer_3UgJ · 2024-10-31

**Soundness:** 3
**Presentation:** 3
**Contribution:** 3
**Rating:** 6
**Confidence:** 4

**Summary:**

This paper introduces the Self-learning Compositional Representation method for zero-shot Chinese Character Recognition, which autonomously learns slots from data. It utilizes a pretraining-inference paradigm and Character Slot Attention to effectively extract relevant components for recognition. Experiments conducted on handwritten and printed Chinese character datasets demonstrate the effectiveness of the proposed method.

**Strengths:**

1) Inspired by object-centric representation learning, this paper proposes a new Self-learning Compositional Representation method. Unlike previous approaches based on radicals or strokes, this method enables the automatic learning of slots from Chinese characters, effectively addressing the challenges of zero-shot Chinese Character Recognition.

2) This paper utilizes Character Slot Attention to map each character image to a group of slots, allowing the model to effectively recognize unseen characters by comparing the slots between input and example images.

3) This paper conducts experiments on handwritten and printed Chinese characters datasets, demonstrating the effectiveness of the proposed method. Visualization experiments further demonstrate that the learned slots effectively represent the different components of the characters.

**Weaknesses:**

1) To further validate the superiority and scalability of the proposed method, we recommend conducting Chinese character recognition experiments in a non-zero-shot setting.

2) The experimental results indicate that as the size of the training set increases, the performance of the model experiences a slight decrease. It might be beneficial for the paper to explore and provide further insights into this phenomenon.

3) The paper notes that models trained on Chinese characters have shown promising performance in Korean and Japanese. However, it would be beneficial to include comparative experiments with other methods to further demonstrate its superiority.

4) Minor issues. In Eq.7, \Lambda^k is not defined or explained. Furthermore, the number of slots on the x-axis in Figure 5 should be expressed in integer units.

**Questions:**

1) Can the proposed method maintain its superior performance in a non-zero-shot setting?

2) Could the model's performance further decline if the size of the training set is increased, and has this possibility been explored in the paper?

3) I am curious whether the proposed method can be extended to Chinese text recognition.

---

### Official Review · Reviewer_g9bJ · 2024-11-01

**Soundness:** 2
**Presentation:** 2
**Contribution:** 2
**Rating:** 3
**Confidence:** 4

**Summary:**

In this paper, the authors introduce a self-learning compositional representation method for zero-shot Chinese character recognition. In this work, the authors leverage a character slot attention module to learn the compositional representation for each character which the training target is to align the  DenseRAN features, therefore it is self-learning. However, the IDS seem also needed in the training, then the dataset also needs some additional label information.  I am not sure if this IDS is removed, what is the recognition accuracy in the experiments.

**Strengths:**

This paper propose a self-learning compositional representation method, which shows the effectiveness in the experiments.

**Weaknesses:**

In the abstract, the authors claim the issues of radical based method and stroke based methods, however, I do not find enough analysis about this in the experiments, I hope the authors can compare and highlight these points in the experiments.

This paper follows the idea from object-centric representation, and leverage the slot-attention module in the zero-shot Chinese character recognition. Hence, I do not find any specific novelty in this area.

The claim that this learned structure of characters in an interpretable way, however, it seems the method can only detect different slots but no the combination order. Hence, how to know the structure of this character is not clear.

The experiments seem show the effectiveness of the method, but more details and analysis should add, like the following questions.

The paper is not written very clear, like the following questions.

**Questions:**

1. The IDS seem also needed in the training, then the dataset also needs some additional label information.  I am not sure if this IDS is removed, what is the recognition accuracy in the experiments.

2. According to my understanding, the reference samples are needed. I do not know what are the reference samples for the zero-shot recognition in the test?

3. Although the authors claim this is auto-learning compositional representation of characters, it seems this is just a normal attention block, so why this slot attention module can learn radical-similar information.

4. In line 322, the authors introduce that if all input images share the same initial states of slots, then acquired components will tend to have the same order. I can not understand this, more explanation are welcome. In case of different order, how to process?

5. In the Table 1, I find larger number of characters, the accuracy is lower for HWDB, but this is opposite for printed, why?

---

### Official Review · Reviewer_bRhH · 2024-11-03

**Soundness:** 2
**Presentation:** 2
**Contribution:** 2
**Rating:** 5
**Confidence:** 5

**Summary:**

This paper introduces SCR-CCR, a method for zero-shot Chinese character recognition that automatically learns compositional components from data. It uses a pretraining-inference approach to recognize characters. Experimental results show that SCR-CCR outperforms existing methods and effectively generalizes to Japanese and Korean characters.

**Strengths:**

1.	SCR-CCR introduces a self-learning compositional representation method that automatically parses compositional objects (slots) from Chinese characters.
2.	The proposed method employs a two-stage process where an encoder, decoder, and ChSA are trained to reconstruct features from character images in the pretraining stage, enabling effective recognition in the inference stage.
3.	SCR-CCR demonstrates the better improvements over existing methods.

**Weaknesses:**

1.	The authors conduct zero-shot CCR experiments using the 500-2755 training dataset. However, it raises the question of whether training on all existing labeled Chinese character data could yield better results in a true zero-shot scenario. If traditional radical-based, stroke-based, or character-based methods perform well in such cases, is the introduction of an object-centric approach necessary? This is a significant setting issue in the current zero-shot CCR field, as genuine zero-shot capability should involve fully utilizing available open-source data and then addressing the unseen scenarios.
2.	The motivation behind the paper is questionable. Both radical-based and stroke-based methods are forms of object-based approaches. While the proposed object-based method allows the model to learn to extract objects autonomously, it is unclear whether this method is truly superior and this motivation is unexplainable.
3.	During the zero-shot inference phase, are the example images printed or of other seen styles (seen during the pretraining stage)? In Figure 2, it appears that handwritten characters of the same style as test image are used.
4.	A critical concern is that the innovation in the proposed method rely on slot attention. Would a standard attention mechanism achieve similar or even better results? This aspect is crucial for an ablation study, as the interpretability of the current object-centric learning paradigm is relatively poor.
5.	In the HWDB experiments presented in Table 1, the performance decreases as the amount of training data increases. This result raises doubts about the validity of the experiment and the method.
6.	Why is the CCR-CLIP performance in this table significantly lower than what is reported in the original paper, which achieved results around 90?
7.	In Table 3, comparisons with state-of-the-art methods from Japanese and Korean studies should be included, especially considering the low accuracy for Korean. Additionally, it needs to be demonstrated whether the proposed method is effective solely for Chinese or also achieves optimal results in other languages.

**Questions:**

As shown in the part of Weakness.

---

### Note · Authors · 2024-11-15

I have read and agree with the venue's withdrawal policy on behalf of myself and my co-authors.